# Mechanism of Arrhythmogenesis Driven by Early After Depolarizations in Cardiac Tissue

**Jack Stein[1], D'Artagnan Greene[1], Flavio Fenton[2], Yohannes Shiferaw [1]***

**1** Department of Physics and Astronomy, California State University, Northridge, California, United States of America, **2** Department of Physics, Georgia Institute of Technology, Atlanta, Georgia, United States of America

* yshiferaw@csun.edu

## Abstract

Early-after depolarizations (EADs) are changes in the action potential plateau that can lead to cardiac arrhythmia. At the cellular level, these oscillations are irregular and change from beat to beat due to the sensitivity of voltage repolarization to sub-cellular stochastic processes. However, the behavior of EADs in tissue, where cells are strongly coupled by gap junctions, is less understood. In this study, we develop a computational model of EADs caused by a reduction in the rate of calcium-induced inactivation of the L-type calcium channel. We find that, as inactivation decreases EADs occur with durations varying randomly from beat to beat. In cardiac tissue, however, gap junction coupling between cells dampens these fluctuations, and it is unclear what dictates the formation of EADs. In this study we show that EADs in cardiac tissue can be modeled by the deterministic limit of a stochastic single-cell model. Analysis of this deterministic model reveals that EADs emerge in tissue after an abrupt transition to alternans, where large populations of cells suddenly synchronize, causing EADs on every other beat. We analyze this transition and show that it is due to a discontinuous bifurcation that leads to a large change in the action potential duration in response to very small changes in pacing rate. We further demonstrate that this transition is highly arrhythmogenic, as the sudden onset of EADs on alternate beats in cardiac tissue promotes conduction block and reentry. Our results highlight the importance of EAD alternans in arrhythmogenesis and suggests that ectopic beats may not be required.

**Data availability statement:** The main code used to generate the data in this study is publicly available at GitHub: https://github.com/yshiferaw/ventricular-cell-model-with-stochastic-calcium. This repository contains all information required to reproduce the main results of the paper.

**Funding:** This work was supported by the National Institute of General Medical Sciences (Award Number: 1R16GM153647-01 to YS) and the National Science Foundation (Award Number: 2320846 to YS). FF was supported by the National Institutes of Health (Award Number: 2R01HL143450-05A1). The funders had no role in study design, data collection and analysis, decision to publish, or preparation of the manuscript. No authors received a salary from any of these funding sources.

**Competing interests:** The authors have declared that no competing interests exist

## Author summary

Early afterdepolarizations (EADs) are voltage oscillations that arise during a cardiac cell's recovery phase. When they occur in large groups of cells, EADs can cause cardiac arrhythmias. Our study uses both mathematical and computational methods to investigate how EADs emerge within cardiac tissue and why they can lead to dangerous arrhythmias. We show that when the mechanism that normally inactivates calcium channels is weakened, EADs appear more frequently and vary from one heartbeat to another. In a single cell, these fluctuations look random. However, when cells are connected in tissue, the coupling between neighbors reduces random variation and makes EAD behavior more predictable. By analyzing a simplified model of calcium dynamics and electrical activity, we find that tissue-scale EADs tend to develop suddenly when the heart's pacing rate crosses a specific threshold. At that point, EADs occur on alternate beats, creating large voltage differences across neighboring tissue. These differences can trigger electrical block, leading to reentrant circuits and possibly life-threatening arrhythmias.

## Introduction

Spatial heterogeneities in the action potential duration (APD) are dangerous because they promote wave break. This occurs when an action potential (AP) wavefront fails to propagate in regions of cardiac tissue that have not fully recovered from a previous excitation. When wave break occurs in a localized area, it can lead to reentry and fibrillation [1]. Additionally, heterogeneities in action potentials can result in after-depolarizations, which may cause premature ventricular contractions (PVCs) which can propagate and induce reentry [2]. Also, early-after depolarizations (EADs) [3–5] are a key contributor to dynamically induced APD heterogeneity. EADs are unusually prolonged action potentials caused by the failure of repolarizing currents to overcome inward currents that prolong the action potential (AP). These prolongations typically occur during phase 2 of the AP and can be caused by an increased L-type calcium current (LCC) or a decrease in potassium currents that control cell repolarization. EADs are characterized by the presence of one or more upstrokes during the AP. EADs are particularly dangerous because they occur randomly in populations of cells, thereby driving APD heterogeneities in cardiac tissue [6–8]. Thus, EADs are often associated with arrhythmias such as Torsade de Pointes, long QT syndromes and heart failure [9–11].

Several studies have investigated the underlying mechanisms of EADs in cardiac cells [12]. In particular, Tran et al. [13] showed that EADs are due to oscillations caused by a Hopf bifurcation that occurs during the AP plateau. Also, Bertran et al. [14] applied canard theory to show that EADs occur when the Ca current is enlarged and is caused by the LCC window current. More recently, Wang et al. [15] identified four distinct mechanisms for EADs, driven by voltage dynamics, calcium dynamics, or their interactions. Also, Huang et al. [16] conducted an extensive study on the

temporal characteristics of EADs, demonstrating that they exhibit strong beat-to-beat fluctuations. This is because the membrane voltage becomes extremely sensitive to ion currents that maintain the AP plateau, amplifying the inherent stochasticity present at the subcellular level. In cardiac tissue, this stochasticity is substantially dampened due to the electrical coupling between cells via gap junction channels, which effectively average the voltage over large populations of cells in tissue. However, it is not well understood how EADs in tissue differ from those observed at the single-cell level. In an insightful study, Sato et al. [17] demonstrated that EADs in tissue are highly arrhythmogenic, inducing stark APD heterogeneities with complex spatiotemporal dynamics. In a subsequent study, Sato et al. [18] analyzed the sources of these heterogeneities and found that they arise from a combination of inherent cellular stochasticity and dynamical chaos due to the system's nonlinearity. These findings indicate that EADs in tissue are both highly nonlinear and stochastic, and both factors may play a key role in the formation of cardiac arrhythmias.

In this study, we use a phenomenological model of a ventricular cell to investigate the properties of EADs in both isolated cells and electrically coupled tissue. The model captures subcellular calcium (Ca) release by tracking the stochastic recruitment and termination of Ca sparks. This stochastic behavior is linked to Ca-sensitive membrane channels, resulting in beat-to-beat variability in the APD. Our findings suggest that, to replicate the experimentally observed APD variations in guinea pig myocytes [19], approximately 4,000 ryanodine receptor (RyR) clusters must be available for Ca spark recruitment. Using this model, we study the formation of EADs in electrically coupled tissue and find that APD fluctuations are significantly dampened compared to isolated cells. In this context, EADs are well described by the deterministic limit of the subcellular stochastic model. Our analysis of this model shows that EADs in cardiac tissue manifest as APD alternans, where the long APD corresponds to an AP with an EAD followed by a short AP with no EAD. A detailed examination of this deterministic model reveals that the transition to EAD alternans occurs via a subcritical pitchfork bifurcation. This transition is discontinuous, so that infinitesimal changes in pacing frequency can lead to large changes in the APD response. Moreover, we show that the presence of EADs in cardiac tissue makes the system highly susceptible to conduction block and reentry. Thus, our results suggest that EAD alternans is a key mechanism for arrhythmogenesis, and ectopic beats are not necessary for the initiation of arrhythmic events.

## Methods

### Modeling Ca spark recruitment in ventricular myocytes

In this study we will develop a phenomenological model of Ca cycling in a ventricular cell which incorporates stochastic dynamics. This model is based on a previously developed model of an atrial cell [20,21] and has been modified for both ventricular cell geometry and electrophysiology. In our approach, we model the stochastic recruitment and extinguishing of the population of Ca sparks within the cell. Since Ca sparks occur randomly, the number of Ca sparks in the cell constitute the primary source of noise in our model. In this approach we separate RyR clusters in the cell into two groups: (i) A population of RyR clusters, referred to as junctional clusters (J), which are in close proximity to LCC channels on the cell membrane. (ii) Non-junctional RyR clusters (NJ) in the cell interior which are far from LCC channels. Under normal conditions Ca release in the cell is due to Ca sparks recruited at J junctions that are triggered by the nearby LCC channel openings. In contrast, NJ junctions are far from LCCs and do not respond to LCC openings but can release Ca if the local Ca concentration rises independently of LCC openings. The distinction between J and NJ clusters is crucial, as the population of RyR clusters activated by LCCs is typically smaller than the total number of clusters in the cell. This distinction is relevant in this study since the fluctuations in Ca spark recruitment depends on the number of clusters that can be activated by LCC openings.

In ventricular myocytes J clusters are distributed uniformly in the cell since t-tubules penetrate into the cell volume and distribute LCCs inside the cell interior [22,23]. In contrast, in atrial myocytes J clusters are distributed mostly on the cell periphery, and Ca release in the interior can only occur due to propagating Ca waves which are due to Ca sparks at NJ clusters [24–26]. In this study we will not consider the effect of wave propagation, although this effect can be added to our

computational framework. To proceed, we will denote $n_b(t)$ as the number of J clusters at which Ca is being released due to a Ca spark at time $t$. Following our previous work [20] we model the change in the number of sparks according to the reaction scheme

$$0 \underset{\beta_b}{\overset{\alpha_b}{\rightleftharpoons}} 1,$$

(1)

where $\alpha_b$ is the rate of Ca spark recruitment at J sites, and where $\beta_b$ is the rate at which these sparks are extinguished. During a small time increment $\Delta t$ the number of Ca sparks in the cell evolves according to

$$n_b(t + \Delta t) = n_b(t) + \Delta n_b^+ - \Delta n_b^-,$$

(2)

where

$$\Delta n_b^+ = B\left(\alpha_b \Delta t, N_b - n_b\right),$$

(3)

$$\Delta n_b^- = B\left(\beta_b \Delta t, n_b\right).$$

(4)

Here, $N_b$ denotes the total number of J clusters, and $B(p, n)$ is a random number picked from a binomial distribution with success probability $p$ and number of trials $n$.

## Cell compartmentalization and Ca cycling

To model Ca dynamics in the cell, we constructed phenomenological equations representing the average Ca concentrations near J and NJ clusters. To develop these equations, we first partition the cell interior into 4 regions which are illustrated in Fig 1A. These volumes are: (i) The total cytosolic volume in the vicinity of J clusters with average free Ca concentration $c_b$ and total volume $v_b$. (ii) The total cytosolic volume in the vicinity of NJ clusters with average free Ca concentration $c_i$ and total volume $v_i$. (iii) The total SR volume in the vicinity of J clusters with an average concentration of $c_{srb}$ and volume $v_{srb}$. (iv) The total SR volume in the vicinity of NJ clusters with an average concentration of $c_{sri}$ and volume $v_{sri}$. Once the number of Ca sparks is computed using equations 2–4 we set the current flux from the SR into the cytosol as

$$J_r^b = g_b c_{srb} p_b(t)$$

(5)

where $p_b(t) = n_b(t)/N_b$ is the fraction of J junctions at which a spark occurs, and where $g_b$ is the effective conductance of RyR clusters in the cell. The other Ca fluxes linking these compartments are illustrated in Fig 1B, and the fluxes in Fig 1A are defined in Table 1. All details of our model construction are given in S1 Text.

## The L-type Ca current

To model the LCC current we will apply a Markovian model, due to Mahajan et al [27], that is based on Rabbit ventricular data. The Markov state diagram introduced in that study, and further developed for population based models [20], is shown in Fig 1B. Briefly, we have split our population of LCCs into two groups of Markov states. The first group, which we will refer to as the "spark off" group, is shown inside the dashed box (Fig 1B) and represents LCCs that are far from active sparks. The Markov states describing these channels consist of 2 closed states ($C_1$, $C_2$), one open state ($O$), and 2 inactive states ($I_1$, $I_2$), with red arrows denoting Ca dependent rates. Since these channels are not in the vicinity of a Ca spark

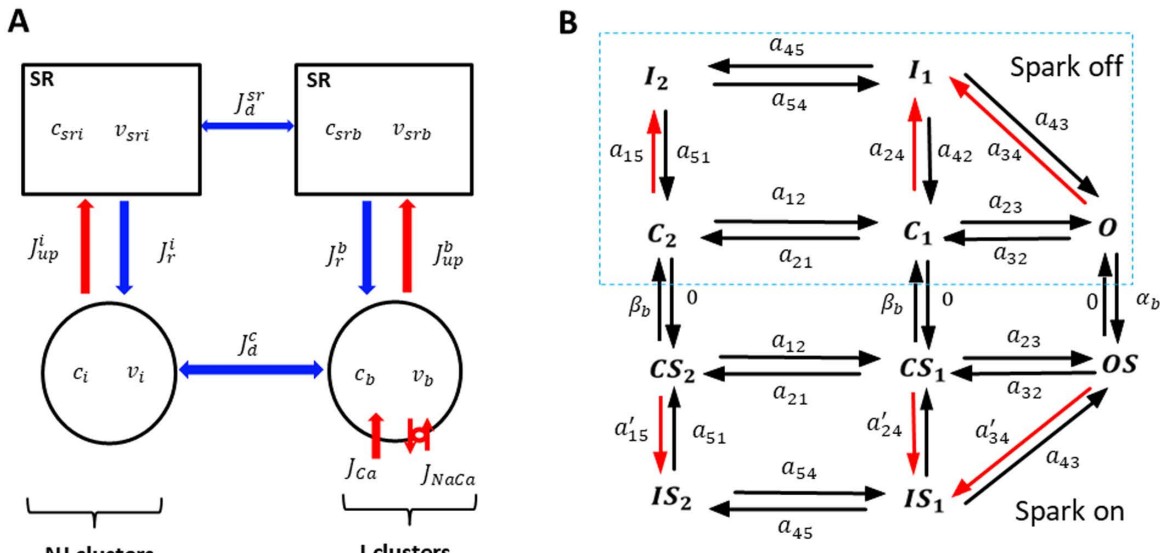

**Fig 1. Phenomenological model of Ca spark recruitment in ventricular myocytes.** (A) Compartment architecture. The cell is divided into 4 compartments which represent the average Ca concentration in the SR and cytosol in the vicinity of J and NJ clusters. The concentrations $c_b$ and $c_{srb}$ denote the average concentration in the cytosol and SR in the vicinity of J clusters, while $c_i$ and $c_{sri}$ denote the concentrations near NJ clusters. The total SR volume near J and NJ clusters is denoted as $v_{srb}$ and $v_{sri}$ respectively, while the total cytosolic volume is denoted as $v_b$ and $v_i$. The currents linking these two spaces are the Ca release $J_r^b$ due to a population of sparks at J clusters, and the uptake pump current $J_{up}^b$ which replenishes the SR volume. Similarly, Ca release and uptake from the SR volume near NJ clusters is denoted by $J_r^i$ and $J_{up}^i$ respectively. The currents $J_d^{sr}$ and $J_d^c$ are averaged diffusive currents proportional to the concentration difference between the internal cell volumes. (B) Markovian model of the LCC current. Markov states in dashed box (spark off) represent channels facing low Ca $(0.1\mu M)$, while channels outside box (spark on) face a Ca spark with high local Ca $(100\mu M)$. The terms $a_{ij}$ denote the transition rates between the corresponding states. Red (black) arrows indicate Ca (voltage) dependent transition rates.

**Table 1. Description of Ca fluxes in phenomenological model.**

| Flux | Description |
| --- | --- |
| $J_r^b$ | Total RyR flux from J clusters to the cytosolic space in the cell interior. |
| $J_r^i$ | Total RyR flux from NJ clusters to cytosolic space in the cell interior. |
| $J_{up}^b$ | Total uptake flux from the cytosol near J sites into the SR. |
| $J_{up}^i$ | Total uptake flux from cytosol near NJ sites to the SR. |
| $J_d^{sr}$ | Total diffusive flux from SR volumes near J sites to NJ sites. |
| $J_d^c$ | Total diffusive flux from cytosol near J sites to NJ sites. |
| $J_{Ca}$ | Total LCC flux into the cell. |
| $J_{NaCa}$ | Total Sodium-Calcium exchanger current into the cell. |

their Ca dependent transition rates depend only on the diastolic Ca concentration in the cell. On the other hand, LCCs that are in the same junctional space of active sparks (spark on) are described by Markov states ($CS_1$, $CS_2$, $OS$, $IS_1$, $IS_2$) which are regulated by a local Ca concentration that is substantially larger. These channels undergo much faster Ca induced inactivation. The transition rates between groups of channels are then modelled by letting open LCCs in the "spark off" group transition to the "spark on" group at a rate that is the spark recruitment rate $\alpha_b$. Similarly, the reverse transition will

be set to the rate that sparks extinguish $\beta_b$. Detailed model parameters describing the channel transitions rates are given in S1 Text.

The transition rates in the LCC model are derived from experimental data established in the Mahajan model. This model was constructed by fitting the LCC current using both barium (Ba) and calcium (Ca) as charge carriers. This approach separated the effect of Ca and voltage dependent inactivation, which is controlled by the transition rates $a_{24}$ and $a_{34}$ (Fig 1B) for LCCs with no local spark, and $a'_{24}$ and $a'_{34}$ for channels in the vicinity of a Ca spark. These rates are taken to have the phenomenological form

$$a_{24} = a^o_{24} + A_{Ca}F_{Ca}\left(c_b\right),$$

(6)

$$a_{34} = a^o_{34} + A_{Ca}F_{Ca}\left(c_b\right),$$

(7)

where $a^o_{24}$ and $a^o_{34}$ determine the rate of inactivation that is independent of Ca. The Ca dependence of inactivation is modelled as a function of the concentration near J clusters, and is given by

$$F_{Ca}\left(c_b\right) = \frac{1}{1+\left(\frac{c_{th}}{c_b}\right)^2},$$

(8)

where $c_{th}$ is the threshold for Ca induced inactivation. Also, the parameter $A_{Ca}$ controls the relative strength of the Ca dependent and Ca independent inactivation rates. In our approach $a'_{24}$ and $a'_{34}$ are taken to have the same form, the main difference being that the Ca concentration $c_b$ in the vicinity of a Ca spark is large so that $c_b \gg c_{th}$ and we set $F_{Ca}\left(c_b\right) = 1$.

### Modeling EADs

EADs have been extensively studied both computationally and experimentally, and it is well established that they can be induced by increasing the LCC [6]. Experimentally, this can be achieved by using a calcium channel agonist such as Bay K. EADs are also observed in congenital long QT (LQT) syndrome [9,28,29] and, in some cases, have been linked to mutations in CaM [28,30], which mediates calcium-induced inactivation. To induce EADs in our model we have reduced the component of calcium-induced inactivation of the LCC current by lowering the parameter $A_{Ca}$ in equations 6–7. This change prolongs the AP and induces EADs at pacing cycle lengths (CL) of $CL \sim 500ms$.

### Action potential model

To model cardiac tissue we have integrated our Ca cycling equations with the major ion currents from the Mahajan AP model for the rabbit ventricular myocyte [27]. In particular we incorporate their ion current formulations for the fast sodium current ($I_{Na}$), the rapidly activating delayed rectifier $K^+$ current ($I_{Kr}$), the slowly activating delayed rectifier $K^+$ current ($I_{Ks}$), the inward rectifier $K^+$ current ($I_{K1}$), the transient outward $K^+$ current ($I_{to}$), the $Na^+/K^+$ exchange current ($I_{NaK}$), and finally the sodium-calcium exchanger current ($I_{NaCa}$).

### Simulations of 2D cardiac tissue

In this study we will explore the dynamics of electrical propagation in a 2D tissue of cells described by our phenomenological model. To model electrical propagation, we apply the cable equation

$$\frac{\partial V}{\partial t} = -\frac{I_{ion}}{C_m} + D_V\left(\frac{\partial^2 V}{\partial x^2} + \frac{\partial^2 V}{\partial y^2}\right)$$

(9)

where $C_m = 1\mu F/cm^2$ is the membrane capacitance, $D_V = 1 \times 10^{-4} cm^2/ms$ is the effective voltage diffusion coefficient, and where $I_{ion}$ is the total transmembrane current. To numerically solve this partial differential equation, we use an operator-splitting scheme [31], that separates the ionic current (reaction) term from the diffusive (spatial coupling) term, allowing stable and efficient time integration. The tissue is discretized on a uniform square grid with a spatial resolution $\Delta x = 0.015 cm$. Time stepping is adaptive, using a variable time step between $\Delta t = 0.01 ms$ during the AP upstroke, and $\Delta t = 0.1 ms$ during repolarization. Using these parameters the conduction velocity on a one dimensional cable is $\sim 10 cm/s$. In this study we examine two main stimulation protocols to investigate cardiac tissue. First, we consider a uniformly excited tissue by applying a short, $2ms$ transmembrane current pulse that depolarizes every cell simultaneously, enabling us to assess the homogeneous response of cardiac tissue. In a second scenario, we investigate planar wave propagation across larger tissue of $100 \times 100$ cells, where the leftmost 10 cells are initially excited. In this manner we can study the properties of planar wave propagation in 2D.

## Results

### Single cell model calibration and fluctuations of the APD

Our ventricular cell model exhibits fluctuations in the voltage time course, which are due to the stochastic recruitment and extinguishing of Ca sparks. The coupling between subcellular stochasticity and the voltage across the cell membrane is mainly driven by the Ca dependence of the sodium-calcium exchange current ($I_{NaCa}$). Since Ca sparks are recruited only from J clusters the beat-to-beat fluctuations in the total number of sparks recruited will depend on the number of available clusters $N_b$. To determine $N_b$ we will rely on the experimental work of Zamboni et al. [19] who measured the beat-to-beat fluctuations of the APD of isolated guinea pig myocytes paced at a steady state cycle length ($CL$). Specifically, they measured the APD at 90% repolarization ($APD_{90}$) and found that the average over 10 beats, after pacing to steady state (2 minutes at $CL = 500ms$), was $\langle APD_{90}\rangle = 342.8ms$ with a standard deviation $\sigma = 10.4ms$. To characterize the magnitude of the fluctuations with respect to the APD they computed the coefficient of variability $c_v = \sigma/\langle APD\rangle$. This measurement was made on 132 myocytes and they obtained $c_v = 2.3 \pm 0.9$ %. To align our model with these findings we have adjusted our number of J clusters ($N_b$) so that our measured $c_v$ at steady state is within the experimental range. In our computational model we find that $N_b \sim 4000$ reproduces the correct order of magnitude of statistical fluctuations observed experimentally. In Fig 2A we show the voltage time course $V(t)$ using $N_b = 4000$. The cell model has been paced to steady state for 50 beats at $CL = 500ms$, and $V(t)$ for the final 20 beats is superimposed on the same graph. In Fig 2B we show the histogram

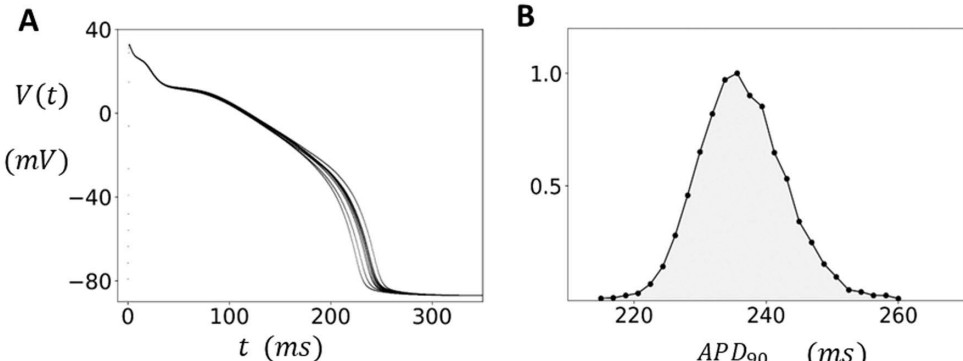

**Fig 2. APD fluctuations in the absence of EADs.** (A) Beat-to-beat voltage fluctuations. The cell is paced at $CL = 500ms$ to steady state (50 beats), and the voltage of the last ten beats is overlayed on the same graph. The number of J clusters is $N_b = 4000$. (B) Distribution of $APD_{90}$. Histogram is constructed by measuring the APD at $90\%$ repolarization over 15000 beats. The distribution is normalized so that the maximum value is 1.

of $APD_{90}$ using a long simulation of 15,000 beats. In this case we find that the average $\langle APD_{90} \rangle = 237ms$ with a standard deviation of $\sigma = 6.3ms$ and $c_v = 2.7\%$. Here, we note that there are potentially many sources of noise that cause APD fluctuations. Our study suggests that Ca cycling likely contributes a significant component of this noise. However, we mention here that the main findings of this study do not depend on the precise origin of the APD fluctuations. In the discussion section, we will assess the robustness of our main results and demonstrate that they remain valid regardless of the specific source of the underlying noise.

**Statistics of EADs in an isolated cell**

In this section, we explore the statistical fluctuations of the APD in the case where the cell model exhibits EADs. To induce EADs we follow the procedure outlined in the methods section, where EADs are induced by lowering the parameter $A_{Ca}$ in equations 6–7. This reduction increases Ca influx by reducing the component of Ca-induced inactivation, which promotes the development of EADs at longer cycle lengths. When EADs occur, the APD is prolonged and full repolarization does not occur on some beats. Thus, to capture the fluctuations in APD we found it necessary to measure the APD when the voltage crosses a higher threshold of $V_c = -40mV$ during the repolarization phase of the AP. Henceforth, all APDs will be computed using this threshold and will be denoted as $APD_{-40mV}$. In Fig 3A, we plot the measured

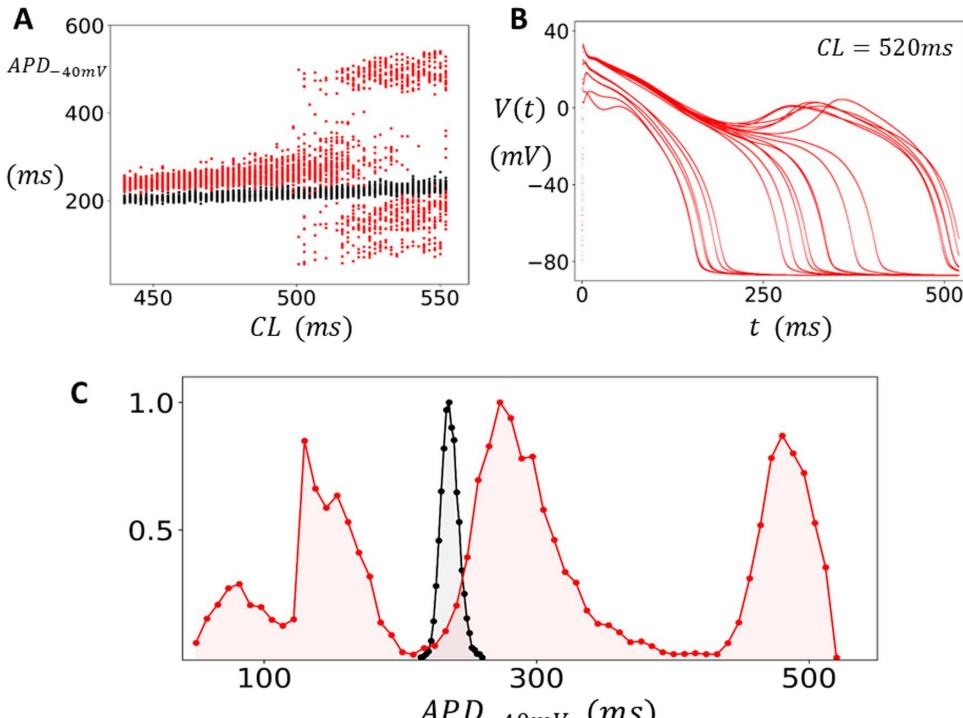

**Fig 3. APD variability in the presence of EADs.** (A) Bifurcation diagram using a dynamic pacing protocol. The cell is paced at a $CL$ for 200 beats and the APD is plotted for the last 50 beats. Upon completion of the 200 beats the $CL$ is increased incrementally and the pacing protocol is repeated. In this manner the $CL$ is swept from $440ms$ to $552ms$. The APD is measured as the time between the AP upstroke and when the voltage crosses $V_c = -40mV$ during repolarization and is denoted as $APD_{-40mV}$. Black points denote normal conditions ($A_{Ca} = 0.15$), and the red points correspond to reduced Ca-induced inactivation ($A_{Ca} = 0.09$). (B) The voltage time course for the last 50 beats when the cell is paced in the EAD regime ($CL = 520ms$). (C) Distribution of $APD_{-40mV}$ measured from 15000 paced beats at $CL = 520ms$. The black filled circles represent the distribution under normal conditions, while the red circles denote EAD conditions. The distributions are normalized so that the maximum value is 1.

$APD_{-40mV}$ over the last 50 beats after pacing to steady state (200 beats), across a range of $CLs$. The black points represent normal Ca-induced inactivation parameters ($A_{Ca} = 0.15$) where EADs do not occur. However, when the component of Ca-induced inactivation is reduced ($A_{Ca} = 0.09$) we find that EADs occur at $CLs$ larger than $CL_c \approx 500ms$ (red points), so that for $CL > CL_c$ EADs occur and large beat-to-beat changes of the $APD_{-40mV}$ is observed. In this simulation we have used a dynamic pacing protocol where the $CL$ is increased incrementally after each round of 200 paced beats. In [Fig 3B], we show an overlay of the voltage time course for the last 50 beats after pacing the cell to steady state at $CL = 520ms$. Here, we observe that the repolarization time course is highly variable, with some beats exhibiting EADs characterized by a notch in the AP, while other beats do not. To quantify the distribution of the APD we have measured the APD for 15000 beats at $CL = 520ms$ and plotted a histogram of the results ([Fig 3C]). At this cycle length we find that the $APD_{-40mV}$ is broadly distributed in the range $\sim 100ms - 500ms$ and is centered around three peaks, which indicates that APD variability increases substantially when EADs occur.

## Statistical fluctuations of the APD in 2D tissue

In cardiac tissue, neighboring cells are coupled by gap junctions, allowing voltage to spread from cell-to-cell. Consequently, the AP measured in cardiac tissue represents the spatial average of the cell population in that vicinity. The number of cells involved depends on the strength of gap junction coupling and the local arrangement of cells. In this section

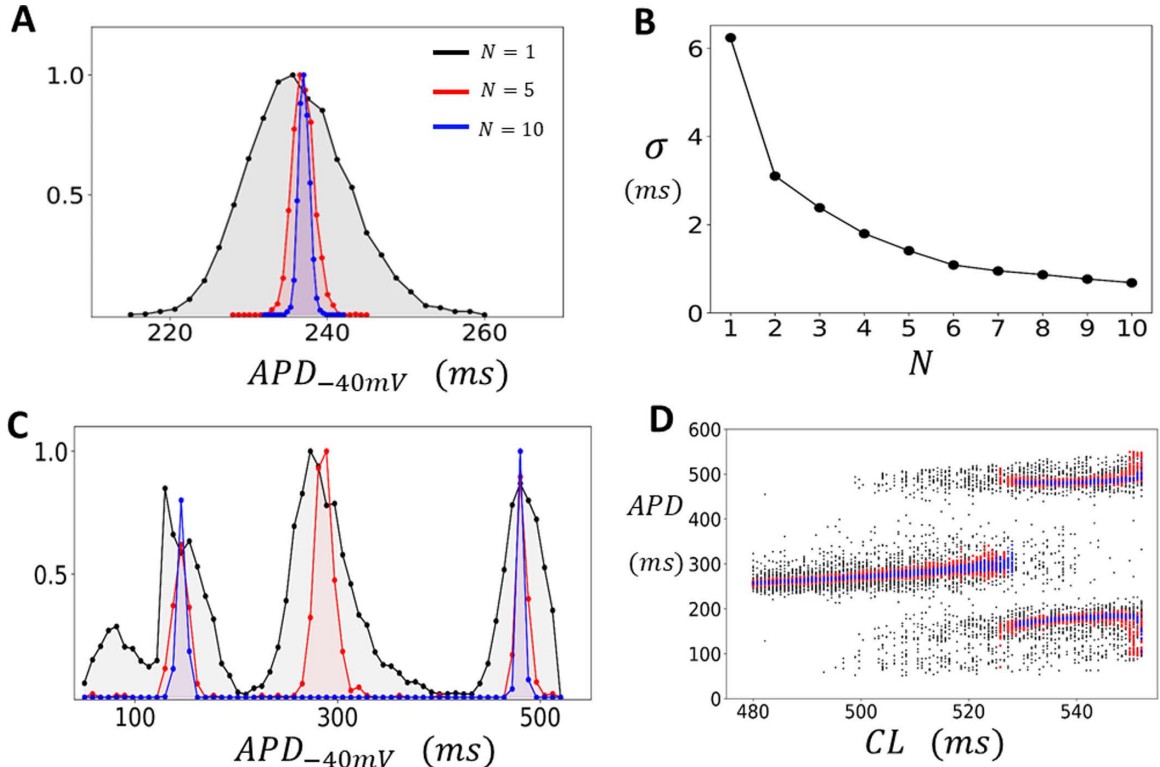

**Fig 4. $APD_{-40mV}$ fluctuations in 2D tissue. (A)** Steady state APD distribution for a 2D tissue of size $N \times N$. The APD is measured at the approximate tissue center at cell ($\lfloor \frac{N}{2} \rfloor, \lfloor \frac{N}{2} \rfloor$). The histogram is constructed by pacing the tissue at $CL = 420ms$ for 5000 beats and recording the APD after an initial transient of 50 beats. The tissue dimension is indicated on the inset. Distributions are normalized so that the maximum is at 1. **(B)** Standard deviation of the APD ($\sigma$) as a function of tissue dimension $N$. **(C)** Steady state APD distribution when the tissue is paced in the EAD regime at a $CL = 520ms$ for 5000 beats **(D)** Bifurcation diagram. The tissue is paced to steady state (200 beats) and the APD is plotted for the last 50 beats. The black, red, and blue points correspond to tissue sizes of $N = 1, 5$ and $10$ respectively.

we apply our computational model to determine how electrical coupling modifies the distribution of the APD in the regime with and without EADs. To model tissue dynamics we solve the cable equation (Eq. 9) in a 2D tissue of size $N \times N$ cells. As a starting point we first analyze how cell coupling modifies the fluctuations of the APD in the absence of EADs. Fig 4A shows the distribution of the $APD_{-40mV}$ for tissue paced uniformly at $CL = 420$ for 5000 beats. In this simulation we have used the model with reduced Ca-induced inactivation ($A_{Ca} = 0.09$) and we measure the APD at the approximate center of our tissue, i.e., at location $\left( \lfloor \frac{N}{2} \rfloor, \lfloor \frac{N}{2} \rfloor \right)$. In this regime EADs do not occur and the APD distribution is shown for tissue sizes of $N = 1$ (black), $N = 5$ (red), and $N = 10$ (blue). As expected, we find that electrotonic coupling substantially reduces the beat-to-beat variations in $APD_{-40mV}$, and this effect becomes more pronounced as the tissue size $N$ is increased. To quantify this effect more systematically, we have computed the standard deviation $\sigma$ of the APD from the data plotted in Fig 4A, and in Fig 4B we have plotted $\sigma$ as a function of tissue dimension $N$. This result demonstrates that the standard deviation of the $APD_{-40mV}$ decreases monotonically as tissue size is increased. In the regime where EADs are observed for an isolated cell ($CL = 520ms$) we find that the $APD_{-40mV}$ distribution is much broader and with a spread that depends on tissue size (Fig 4C). To explore this effect, we have also plotted the bifurcation diagram of the system (Fig 4D) where the tissue is paced to steady state for a range of $CL$. In this case we observe that in the EAD regime the APD is broadly distributed around two or three peaks with a spread that decreases with increasing tissue size.

## Large tissue sizes are governed by the deterministic limit of the stochastic APD model

To understand the distribution of the APD in cardiac tissue we will consider the deterministic limit of our stochastic Ca model. Our motivation is that in cardiac tissue the voltage is averaged over a population of cells so that Ca sparks are effectively recruited from a large number of J clusters. Thus, we can take the $N_b \to \infty$ limit of equations (2–4). In this limit the number of Ca sparks in a population of cells is governed by a deterministic equation given by

$$\frac{dp_b}{dt} = \alpha_b \left( 1 - p_b \right) - \beta_b p_b, \tag{10}$$

where $p_b = n_b / N_b$ is the fraction of J clusters with sparks. To compare the deterministic model predictions with our tissue simulations we first pace different tissue sizes to steady state and measure the average APD ($\langle APD \rangle$) and standard deviation ($\sigma$). In Fig 5A we plot the $\langle APD \rangle$ (black circles) and $\sigma$, represented as an error bar around the average, when the cell is paced at $CL = 420ms$. On the same graph we plot the steady state APD predicted by the deterministic model (red line). Indeed, we find that as the size of the system is made larger the average APD of the stochastic model in tissue converges to the prediction of the deterministic limit. To explore the regime where the system exhibits EADs, in Fig 5B we plot the APD after the deterministic system has been paced to steady state. Specifically, the cell is paced at a given $CL$ for 200 beats, and the APD of the last 50 beats is recorded. The $CL$ is then increased by a small increment, and the protocol is repeated. This procedure is repeated for $CLs$ in the range $CL_0 = 475ms$ to $CL_1 = 550ms$. Once the $CL_1$ pacing is completed the protocol is repeated, and $CL$ is decreased from $CL_1$ back to $CL_o$ (red filled circles). Indeed, in this simulation we find a transition to alternans (black filled circles) when the CL is increased from $CL_o \to CL_1$. However, when the protocol is repeated in the reverse direction $CL_1 \to CL_o$ we find that the alternans regime continues for a broader range of $CLs$. This result indicates that the steady state APD exhibits a regime of bistability. On the same plot we show the APD for the last 50 beats when the stochastic single cell model (blue dots) is paced to steady state. Here, we observe that the beat-to-beat fluctuations of APD in the single cell case are broadly scattered around the predictions of the deterministic model. Thus, at a finite system size $N$, the stochasticity from subcellular fluctuations is superimposed on the nonlinear dynamics dictated by the deterministic model. In Fig 5C we plot the voltage time course for the single cell stochastic model (black lines), and the deterministic model (red line), when the cell is paced at $CL = 530ms$. Indeed, we find that the single cell displays large beat to beat fluctuations in the APD, while the deterministic model exhibits APD alternans where EADs occur on alternate beats.

## APD fluctuations in uniformly paced cardiac tissue

The findings from the previous section indicate that cardiac tissue exhibits beat-to-beat variations in APD due to a combination of subcellular noise and the nonlinear response of ion currents. This suggests that paced cardiac tissue will display spatial variations in the APD that is caused by these factors. To analyze these spatial patterns, we simulated a strip of cardiac tissue where all cells were paced periodically. Specifically, we paced a $3 \times 100$ array of coupled cells at a cycle length $CL_1$ until steady state was reached (50 beats). We then changed the cycle length to $CL_2$ and continued pacing at this new cycle length for 300 beats. Once a steady state was reached at $CL_2$, we evaluated the spatial distribution of the APD on the tissue strip. In Fig 6A we plot the *APD* as a function of beat number *n* for a cell that is at the center of a tissue that is paced at $CL_1 = 400ms$ and $CL_2 = 420ms$. In Fig 6B we plot the APD as a function of the number of cells $I_x$ along the cable, after 200 (black line) and 201 (red line) beats of pacing at $CL_2$. Here, we see that there are small spatial variations of the APD across the tissue, which are due to the underlying stochasticity of the system. In Fig 6C we show the *APD* vs *n* when the tissue is paced in the bistable regime ($CL_2 = 520ms$). In this case we find that the APD exhibits large intermittent fluctuations where the APD can vary from $100ms$ to $500ms$. In Fig 6D we show the spatial distribution of APD along the cable at beats $200$ and $201$. This result demonstrates that in the bistable regime the APD distribution is dynamic and can vary substantially along the cable from one beat to the next. This spatial variation is driven by fluctuation induced

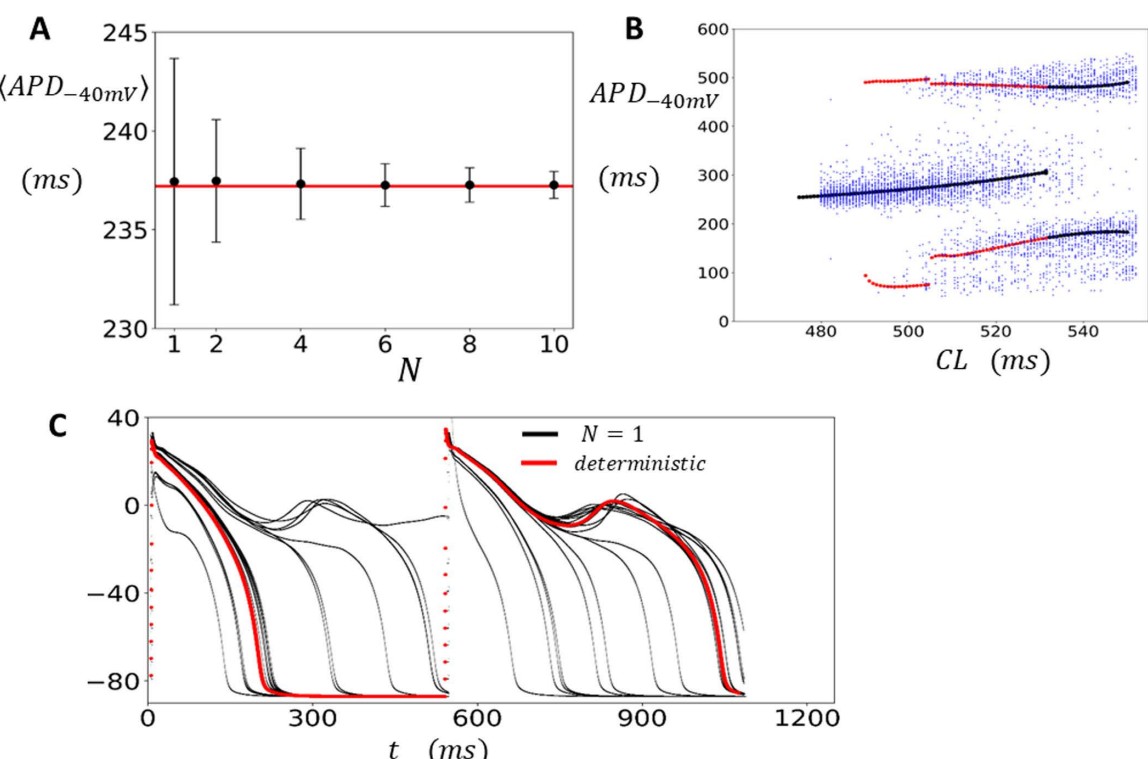

**Fig 5. Deterministic limit of the stochastic model.** (A) Plot of $\langle APD \rangle$ as a function of tissue size *N* (black filed circles). The standard deviation $\sigma$ is plotted as an error bar. The tissue is paced for 5000 beats at $CL = 420ms$. The horizontal red line corresponds to the deterministic limit paced to steady state (50 beats). (B) Bifurcation diagram for the single cell stochastic model (blue filled circles) along with the predictions of the deterministic model (red and black filled circles). The deterministic bifurcation diagram is computed using a dynamic pacing protocol where the *CL* is increased gradually from $CL_o = 475ms$ to $CL_1 = 550ms$ (black points). At each *CL* the cell is paced for 200 beats and the APD for the last 50 beats are plotted. Once $CL_1$ is reached the *CL* is then decreased back to $CL_o$ (red filled circles). (C) The steady state voltage plotted for the last 50 beats, at $CL = 535ms$, for the stochastic (black line) and the deterministic (red line) model.

transitions between the two stable regimes. In Fig 6E and 6F we have paced the cable in the alternans regime where $CL = 530ms$. In this case the APD in tissue alternates from beat-to-beat (Fig 6E) and exhibits a stable pattern of spatially discordant alternans (Fig 6F). These results demonstrate that EADs in tissue induce dynamic spatial heterogeneity of the APD.

## Subcellular stochasticity is highly arrhythmogenic in the EAD regime

In this section, we analyze the effect of EADs on wave propagation in 2D cardiac tissue. Fig 7 shows simulations of a $100 \times 100$ tissue, paced along a 10-cell-wide strip on the left edge. Initially, we pace the tissue for 20 beats at a cycle length (CL) of 420ms, where the cell model does not exhibit EADs. We then switch to a CL of 530 ms, where EADs occur in the deterministic limit. Fig 7 includes snapshots of planar wave propagation at the beat numbers indicated. By the 40th beat, the AP wave back begins to develop pronounced heterogeneities as EADs form in the tissue. These heterogeneities increase until, on the 43rd beat, the AP wavefront breaks and forms a reentrant circuit. This wave break occurred in all simulations runs at CL = 530ms and is likely a robust feature of the system. In the discussion section, we will analyze

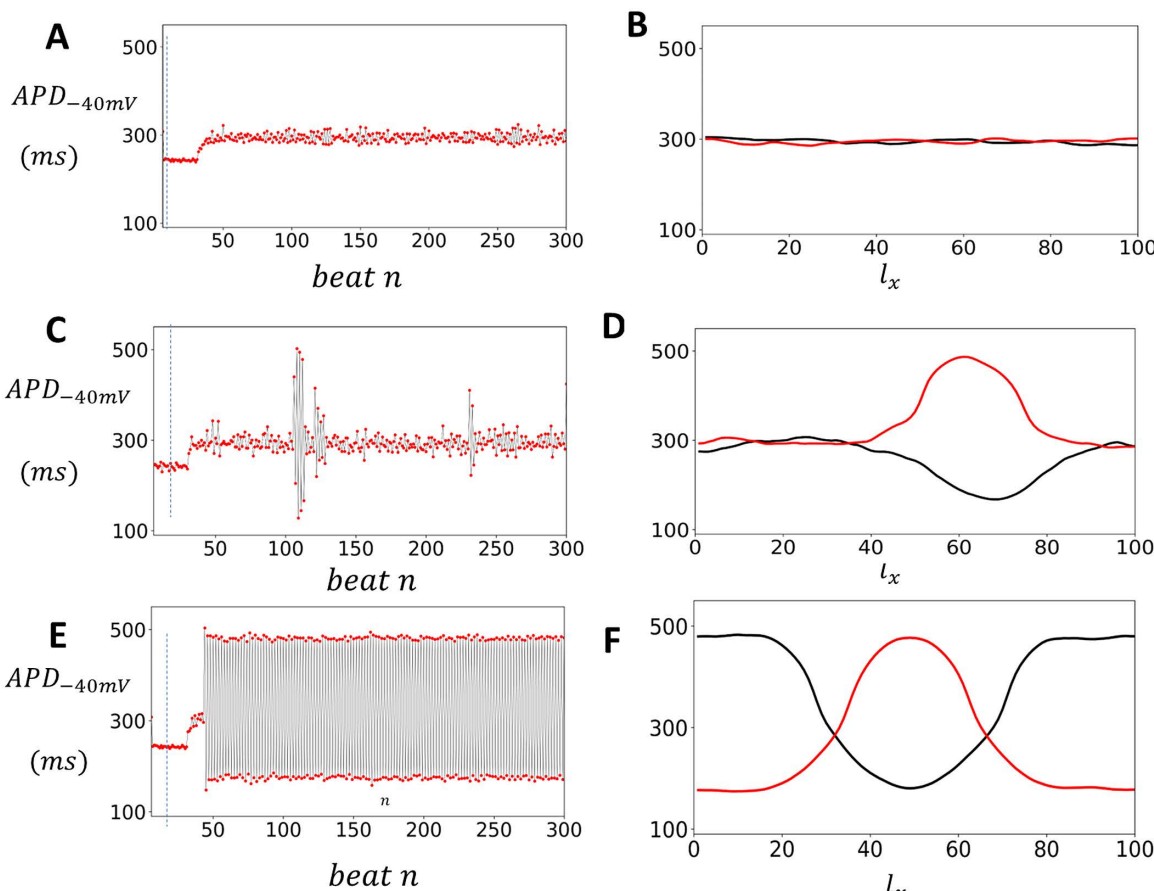

**Fig 6. Spatial patterns and fluctuations in APD in cardiac tissue with EAD.** (A) $APD_{-40mV}$ as a function of beat number $n$ for a cell at the center of a $3 \times 100$ tissue array, paced at $CL_1 = 400ms$ for 30 beats and $CL_2 = 420ms$ for 270 beats. (B) $APD_{-40mV}$ as a function of the number of cells $l_x$ along the cable after 200 (black line) and 201 (red line) beats at $CL_2$. (C) $APD_{-40mV}$ versus $n$ when the tissue is paced in the bistable regime ($CL_2 = 522ms$), displaying large intermittent fluctuations. (D) Spatial distribution of $APD_{-40mV}$ along the cable at beats 200 and 201 in the bistable regime. (E) $APD_{-40mV}$ vs $n$ when the system is paced in the alternans regime ($CL = 530ms$). (F) Spatial distribution of $APD_{-40mV}$ at beats 200 and 201 showing stable spatially discordant alternans.

the mechanism for wave break and show that it is a direct consequence of the spatial heterogeneity caused by EADs in cardiac tissue.

## Discussion

### Statistics of EADs in isolated cells and tissue

In this study, we developed a computational model to describe the formation of EADs resulting from reduced calcium-induced inactivation of the LCC. The model incorporates subcellular stochasticity by accounting for the random activation and termination of calcium sparks within a population of RyR clusters in the cell. Our approach reveals that the magnitude of APD variability during pacing is sensitive to the number of RyR clusters where calcium sparks can potentially occur—specifically, those clusters that can be triggered by openings of the LCC. Our findings suggest that, to match experimental data on APD fluctuations in guinea pig myocytes, approximately 4,000 RyR clusters are required as potential sites for calcium spark initiation. This result aligns with experimental observations indicating that cardiac myocytes contain roughly $10,000-20,000$ RyR clusters [32–34], with only a fraction of these clusters capable of serving as calcium spark nucleation sites[35]. However, it is clear that the number of clusters that can be recruited will depend on the strength of the LCC current, which itself depends on a wide range of physiological factors such as $\beta$-adrenergic stimulation. Thus, while APD fluctuations are dynamic and will have many sources, our study suggests that Ca spark recruitment is likely a significant contributor. Here, we emphasize that in our model, fluctuations originate from stochastic Ca spark recruitment by LCCs which induces beat-to-beat variability in the amount of Ca release. This in turn induces fluctuations in $I_{NaCa}$ which drives APD fluctuations. It should be noted that Ca waves that occur during the AP can potentially drive the formation of EADs.

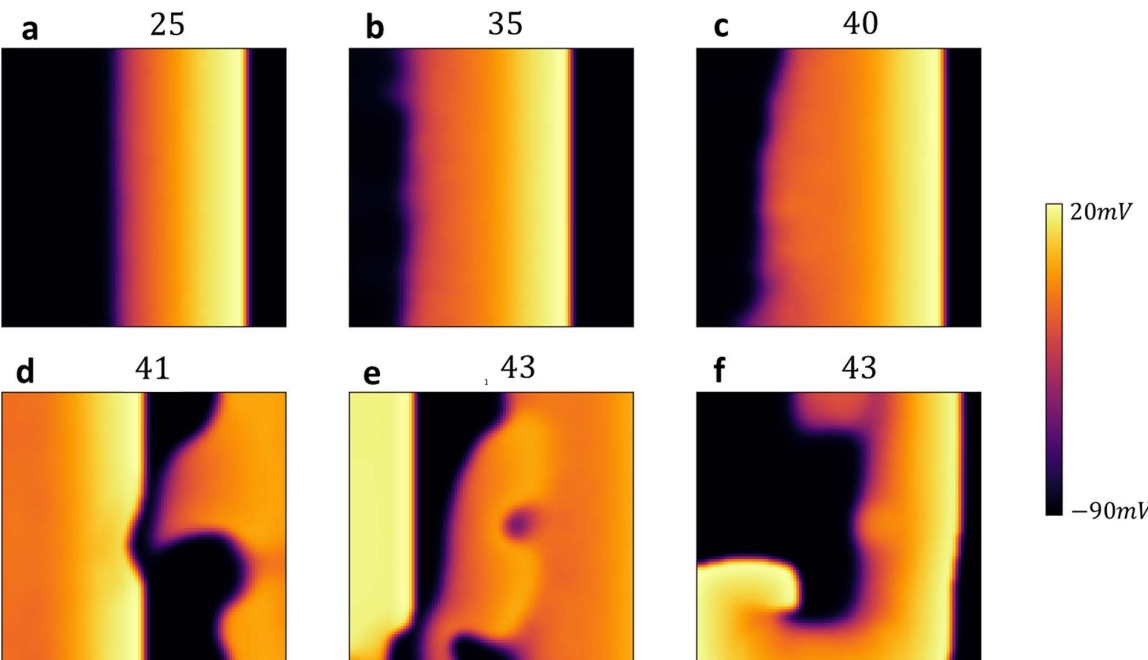

**Fig 7. Wave Propagation in 2D Cardiac Tissue with EADs.** Snapshots of planar wave propagation in a $100 \times 100$ cardiac tissue, paced along a 10-cell-wide strip on the left edge. The tissue is initially paced for 20 beats at $CL = 420ms$, where the cell model does not exhibit alternans. Subsequently, the CL is switched to $530ms$, where EADs occur in the deterministic limit. The snapshots show the development of pronounced heterogeneities in the action potential (AP) wave back by the 40th beat, leading to wavefront break and reentrant circuit formation by the 43rd beat. Snapshot **f** is 400ms after snapshot **e** during the 43rd beat. The full movie is available in S1 Movie.

However, this mechanism is distinct from the mechanism proposed here which depends on EADs induced by the reduction of Ca-induced inactivation of $I_{Ca}$.

When we reduced the extent of calcium-induced inactivation by decreasing the calcium-dependent component, we observed that EADs occurred at slower pacing rates. At these rates, the APD fluctuated by over $400ms$ from beat-to-beat as EADs amplified the subcellular stochasticity. This outcome is expected, since many studies have demonstrated that EADs arise from a nonlinear instability during the AP plateau, which makes the repolarization dynamics highly sensitive to subcellular fluctuations [13,17]. Our results are also consistent with experimental findings, which have shown that EADs are associated with pronounced APD variability. Notably, Li et al. [36] measured EADs over multiple beats in dog ventricular myocytes in heart failure, finding that when these myocytes were paced at 1Hz, the APD varied by up to $\sim 400ms$ during EADs. This observation is consistent with our finding that EADs in isolated cells significantly amplify subcellular fluctuations.

In cardiac tissue, cells are electrically coupled, so that the voltage of each cell will be the spatial average of a large population of neighboring cells. Thus, electrical coupling will reduce the beat-to-beat variability of the APD observed in an isolated cell. To analyze this effect more generally, we note that electrical coupling spreads over a length scale of $l \sim \sqrt{D_V \cdot T}$, where $D_V$ is the effective diffusion coefficient of voltage in tissue, and where $T$ is the pacing period. Setting $T = 500ms$ and using a standard diffusion coefficient of $D_V \sim 10^{-4}cm^2/ms$, we find that $l \sim 0.2cm$. Consequently, in cardiac tissue, the APD will represent the spatial average of the population of cells within a spherical volume of this radius, which is on the order of $10^4$ cells. Now, since the noise strength decreases as $1/\sqrt{N}$, where $N$ is the effective number of recruitable RyR clusters contained in that population of cells, then we have $N \sim 10^7$, since each cell will have roughly $\sim 10^3$ RyR clusters. Thus, in 3D tissue, the subcellular noise strength will be diminished by a factor $\sim \frac{1}{\sqrt{N}} \approx 10^{-4}$, so that the voltage time course will be effectively deterministic. However, in one-dimensional (1D) strands of cardiac tissue, the situation is markedly different. Here, only about 20 cells fall within the same electrotonic length. The reduced number of cells means that stochastic effects become much more significant. As a result, fluctuations in APD due to EADs will be far more pronounced in 1D tissue than in 3D tissue. Therefore, while 3D cardiac tissue exhibits mostly deterministic behavior with minimal influence from noise, 1D cardiac tissue is heavily influenced by stochasticity, leading to substantial variability in the APD due to EADs. Additionally, the effective diffusion coefficient of voltage, $D_V$, is highly sensitive to gap junction coupling, with regions of reduced coupling exhibiting greater fluctuations. This suggests that EAD fluctuations will be particularly pronounced in regions of low gap junction coupling. Moreover, increased fibrosis and connexin remodeling can further exacerbate these effects by disrupting electrical connectivity and enhancing spatial dispersion of repolarization. Finally, we should mention that recent studies have pointed out the crucial role of ephaptic coupling in reducing electrotonic load during repolarization [37,38]. Also, recent work has pointed out the relevance of fractional diffusion to account for the heterogeneous coupling of cardiac cells in tissue [39,40]. These studies imply that the mechanism presented here can potentially be even more pronounced in cardiac tissue. By facilitating local depolarization and altering current flow dynamics, ephaptic interactions could further enhance the conditions that promote EAD formation, particularly in structurally heterogeneous regions. Future studies incorporating these effects will be essential for fully understanding their impact on arrhythmogenesis.

Our simulations confirm that a one-dimensional tissue can exhibit complex spatiotemporal behavior. As shown in Fig 5B, a bistable regime exists where a period-2 solution coexists with a period-1 solution. In this regime, fluctuations between the bistable states lead to intermittent excitations, as depicted in Fig 6C. These fluctuations along the one-dimensional cable generate sharp heterogeneities, with different regions of the tissue shifting between distinct bistable states (Fig 6D). This bistable regime spans approximately from $480ms$ to $530ms$. For pacing rates above $530ms$, in the period-2 regime, the one-dimensional cable evolves toward a spatially discordant pattern. This regime is also arrhythmogenic, as it induces large APD heterogeneities in the tissue, as shown in Fig 6F. Analysis of the spatial patterns reveals that the spatial gradients in the bistable and period-2 regimes are similar. In both cases, the APD can vary dramatically

from beat to beat, with spatial gradients forming over a characteristic length of approximately $l \sim \sqrt{D_V \cdot T} \sim 0.2 cm$. However, in the intermittent regime, spatial variations are transient and do not persist at steady state. Consequently, we expect fluctuating patterns in the tissue during the intermittent regime, whereas in the period-2 regime, stable patterns will emerge. Overall, our study reveals that early afterdepolarizations (EADs) in tissue are highly arrhythmogenic and can drive pronounced spatial heterogeneities, potentially leading to localized conduction block [41,42].

## Nonlinear dynamics and EADs

A crucial finding of this study is that in 2D cardiac tissue, APD fluctuates around values determined by the deterministic limit of the stochastic model. Thus, in order to understand the statistical distribution of the APD, it is necessary to characterize the beat-to-beat dynamics of the deterministic model. To achieve this, we will construct a nonlinear map model of the APD, which will identify the key nonlinear properties driving the instability. Our approach is to determine the functional dependence of the APD on the previous diastolic interval, denoted as $DI$, by computing the S1S2 restitution curve of the model. This approach has been used in previous studies to analyze period doubling bifurcations in cardiac cells [43]. To construct the S1S2 restitution curve, we first pace the cell at the S1 cycle length. Once the cell has reached steady state, we vary the CL of the last paced beat, referred to as the S2 stimulus, and measure the final APD and the preceding $DI$. This procedure is repeated for different S2 durations and in Fig 8A, we plot the final APD versus $DI$. The S1S2 restitution curve for our model indicates that the APD increases gradually as a function of $DI$, until roughly $DI \sim 350ms$, after which it increases abruptly. To explain the shape of the restitution curve, in the inset we show the voltage time course for the last paced beat, corresponding to the four representative points on the restitution curve. This shows that the steep increase of the APD is due to the onset of EADs which induces long APDs when $DI$ approaches a critical value. Here, we note that the same shape of the APD restitution curve has been observed in previous studies where EADs are induced by different ionic mechanisms [18,44]. Thus, this APD restitution shape is not specific to our model, but is expected more generally in ionic models which exhibit long APDs characteristic of EADs. Also, we emphasize that the steepening of the APD restitution curve can occur even without EADs, provided that the APD lengthens as a function of DI, which can occur in the absence of EADs, as in the case of Brugada syndrome [45]. Thus, the mechanism proposed here does not rely on the presence of the secondary upstroke during the AP plateau. To explore the effect of this nonlinearity we have fitted a cubic spline to the restitution curve (Fig 8A, black line) and used this function to construct a nonlinear map of the form

$$A_{n+1} = F(DI_n),$$ (11)

where $A_n$ and $DI_n$ denote the APD and DI at beat $n$, and which is iterated at fixed CL denoted as $T = DI_n + A_n$. In Fig 8B, we present the bifurcation diagram for the nonlinear map, showing a transition from a stable periodic fixed point to a period doubled steady state. To generate the diagram, we iterated the map at fixed values of $T$, incrementally increasing $T$ after reaching steady state, and plotted the steady-state APD (blue circles). Red circles represent data obtained by decreasing $T$ from the period-doubled solution. These results indicate that the period-doubling bifurcation is bistable and that the periodic fixed point loses stability discontinuously. To analyze these features further in Fig 8C we show cobweb diagrams for $T = 550ms$ using two initial conditions: $DI_o = 350ms$ and $DI_o = 450ms$. This result indicates that the period doubled solution corresponds to EADs occuring only on alternate beats. These results are consistent with the bifurcation diagram of the full ionic model, which also exhibited a similar bistable period-doubling bifurcation. Finally, we emphasize that the bifurcation observed here is not unique to our model. A similar bifurcation has been reported in other models of EADs involving different mechanisms. For example, Liu et al. [44] investigated a Long QT syndrome model and found that the transition to EADs occurred through a discontinuous transition to EAD alternans. This suggests that the transition we describe is a generic feature of all EAD models and not confined to a specific cellular mechanism.

Our nonlinear map analysis uncovers several important characteristics of the transition to EAD alternans in the deterministic limit. The first important observation of the transition, in both the nonlinear map and the full ionic model, is that it is discontinuous. To explain this feature, we apply standard methods to analyze how the periodic fixed point of the map bifurcates into the period doubled solution. Our approach is to rewrite Eq. 11 as the single variable map $DI_{n+1} = G(DI_n)$, where $G(DI) = T - F(DI)$. To analyze the period doubled solutions we consider the second iteration of the map, $G_2(DI) = G(G(DI))$, so that a period doubled solution satisfies the condition $DI = G_2(DI)$. In Fig 8D we study the solutions of $G_2(DI) - DI = 0$ as the pacing period $T$ is varied across the transition. The red line shows the function $G_2(DI) - DI$ for

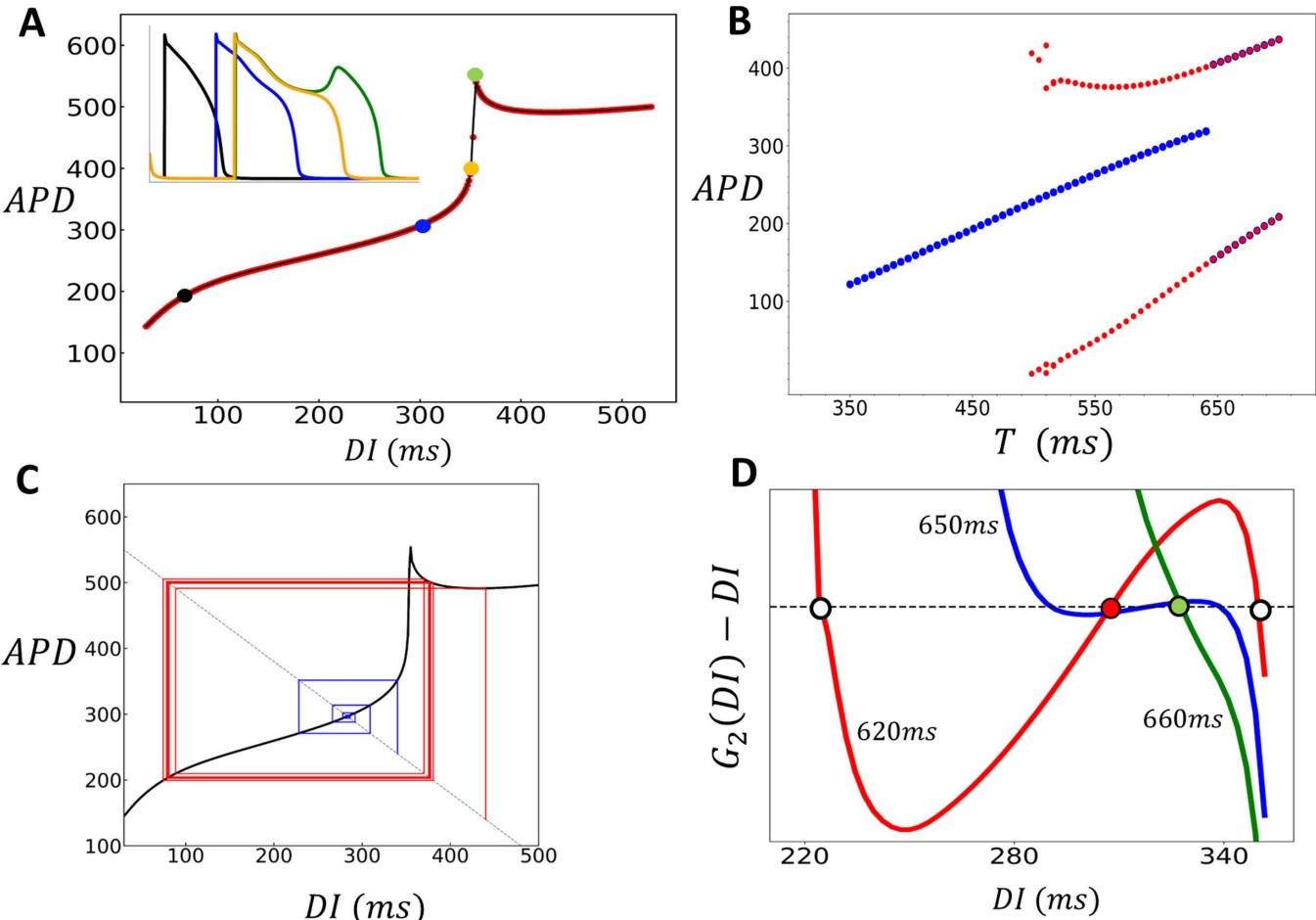

**Fig 8. Analysis of Nonlinear Dynamics and EAD-Induced Bifurcations in Cardiac Tissue.** (A) S1S2 restitution curve. The final APD is plotted against the previous diastolic interval ($DI$) after pacing the cell to steady state (50 beats) at $S1 = 500ms$. The red points are the measured APD for the S2 beat and the black line is a cubic splines fit. The inset shows voltage for the last paced beat at the indicated points on the restitution curve. The black, blue, yellow, and green plots correspond to the colored points on the restitution curve. (B) Period-doubling bifurcation diagram. The presence of EADs leads to a period-doubling bifurcation, resulting in an alternating behavior where EADs occur only on alternate beats. Blue points are computed by sweeping the period from low to high, and the red points are computed by sweeping the period from high to low. (C) A cob-web diagram demonstrating bistability of the nonlinear map where the stable fixed point (blue) coexists with an alternating behavior (red). The pacing period is fixed at $T = 550ms$. (D) Plot of the function $G_2(DI) - DI$, where $G_2(DI)$ is the second iteration of the S1S2 restituion map. Intersections with the dashed line indicate fixed points of the second iteration of the nonlinear map. At $T = 620ms$ (red line) the system posseses two unstable fixed points (white circles) and one stable fixed point (red circle). At $T = 660ms$ (green line) the system only possesses one unstable fixed point (green circle). This transition occurs occurs near $T \approx 650ms$ where the stable fixed point loses its stability via a subcritical pitch fork bifurcation (blue line). Note that this plot only shows the range of $DI$ near the stable fixed point, as the system possesses two other stable fixed points corresponding to the bistable state.

the case $T = 620ms$ which is less than the onset of the period doubling bifurcation at $T_c \sim 650ms$. Here, we see that there are two unstable solutions (white circles), and one stable solutions (red circle). However, above the transition, where $T = 660ms$ (green line), we find that there is only a single stable solution (green circle). Also, we plot the function near onset at $T = 650ms$ (blue line) to see that the two unstable fixed points vanish at the instant where the stable fixed point loses stability. This transition is known as a subcritical pitchfork bifurcation, which is characterized by a discontinuous jump to alternans. This type of bifurcation typically occurs when the system exhibits an approximate reflection symmetry around the stable fixed point. In our system, this symmetry is roughly satisfied due to the shape of the restitution curve near the onset of EADs. This behavior is in sharp contrast to the transcritical bifurcation seen in the standard APD restitution instability to alternans [1], where APD alternans develop gradually without a discontinuous jump.

Our nonlinear analysis of EADs in cardiac tissue reveals several findings worth noting. First, in paced cardiac tissue, EADs appear as alternans, occurring only on every other beat. This occurs because cardiac tissue can be effectively modeled by the deterministic limit of our stochastic model, where EADs arise following a period-doubling bifurcation. Importantly, this alternating behavior is observed only at the tissue level. In isolated cells subcellular noise introduces significant variability in EAD occurrence and a consistent alternans response will not be observed. Second, this bifurcation is discontinuous so that even an infinitesimal change in pacing rate leads to a large change in the APD. This feature causes EADs in tissue to occur abruptly since they emerge after a discontinuous transition. Finally, the transition to EAD alternans is bistable, so that the alternating EAD alternans can coexist with a stable periodic state. This result has important physiological implications. In particular, the onset of EADs will be highly sensitive to pacing rate and tissue heterogeneities. Thus, when tissue becomes prone to EADs they will likely emerge in a highly heterogenous fashion. Also, since the transition is bistable different parts of tissue can exhibit either a periodic or alternating response. These features ensure that when cardiac tissue is paced near the EAD threshold will exhibit steep APD gradients which are highly arrhythmogenic.

### EADs promote conduction block in paced cardiac tissue

Our pacing simulations in 2D cardiac tissue (Fig 7) revealed that when EADs occur, a planar excitation becomes highly unstable, leading to conduction block and re-entry. To better understand this phenomenon, we computed the S1S2 restitution curve in a strip of cardiac tissue, using the same S1S2 pacing protocol as described in the previous section. To generate the restitution curve, we paced the left end of the tissue at a fixed cycle length (CL) and measured the APD at the distal end. We then applied the S1S2 protocol and measured the APD as a function of the preceding $DI$ for the final paced beat. The resulting restitution curve is shown in Fig 9A. This curve is similar to the single cell restitution curve with the difference that when $DI$ falls below approximately $25ms$, the AP fails to propagate, resulting in conduction block. The range where propagation fails is highlighted by the green shaded box in the figure. The presence of a conduction block regime implies that an S2 beat can fail to propagate if delivered bellow a fixed threshold. In Fig 9A, this is illustrated using cobweb diagrams for two different initial conditions, shown in red and blue. When the S2 is applied with an intermediate $DI$ (blue line), the system converges to a stable fixed point. However, when the $DI$ is larger (red line), the electrical excitation fails to propagate, resulting in conduction block. This occurs due to the distinctive shape of the APD restitution curve, which leads to large APDs when EADs are present. As a result, a long APD prevents the tissue from adequately recovering, causing conduction block on the following beat. To further investigate this effect across a range of initial conditions, we used a cubic spline fit to the APD restitution curve and analyzed how different initial conditions evolve over many beats. In Fig 9B we have computed the phase diagram of the system. The y-axis represents the initial diastolic interval $DI_0$, while the x-axis represents the pacing period ($T$). For each initial condition, the nonlinear map was iterated 100 times to determine the system's long-term behavior. In the resulting phase diagram, blue regions indicate a stable period-1 solution, red regions represent a stable period-2 solution, and black regions denote the occurrence of conduction block. This result demonstrates that APD propagation in tissue with EADs is extremely sensitive to small changes in the pacing period. Therefore, in paced cardiac tissue—where the timing of an excitation wavefront is expected to vary—the

system becomes highly prone to wavebreak. It is important to contrast the mechanism proposed here with the usual APD alternans that occur at fast pacing rates. In those cases, the amplitude of APD alternans increases gradually as the cycle length decreases, progressively engaging the steep part of the CV restitution curve at small DIs. In contrast, in our case, the large APD amplitude immediately engages this steep regime, leading to immediate and large beat-to-beat variations in CV when the system is paced in the period-2 regime. Finally, we emphasize that our simulations are conducted in isotropic tissue. In the more realistic case of anisotropic tissue, where transverse and longitudinal coupling differ significantly due to fiber orientation, we expect our results to be highly sensitive to these effects. Weak gap junction coupling in a given direction will amplify fluctuations, making wavebreak tendency strongly dependent on fiber orientation.

Our analysis suggests that EADs in tissue occur via alternans which promotes conduction block and reentry. However, several studies have suggested that EADs can induce arrhythmias via propagating ectopic beats [46,47]. In this scenario an EAD in tissue propagates into nearby refractory tissue, and that excitation proceeds to form reentry. Though this scenario is possible, the conditions for successful propagation of an EAD in cardiac tissue are quite stringent. For instance, Deo et al. [48] analyzed ectopic excitations resulting from EADs in the Purkinje system and showed that these excitations can lead to reentry, but only under precise conditions. Specifically, they identified strict requirements on the timing of the excitation and the electrophysiological state of the substrate for reentry to occur. In contrast, our study of paced cardiac tissue revealed that reentry occurred robustly following the onset of EADs without requiring ectopic excitations. Instead, it was the heterogeneity introduced by EAD alternans that led to conduction block, which then facilitated reentry. To explore this possibility further it is essential to extend our methods to realistic three-dimensional geometries using computational frameworks like openCARP [49]. This approach will allow researchers to account for intricate spatial interactions and electrotonic effects that vary across complex tissue structures, including the Purkinje–myocardial junction, which was recently identified as a critical site for arrhythmogenesis [50]. By analyzing how EAD alternans and conduction block emerge and propagate in these anatomically and functionally diverse regions, we can gain deeper insights into arrhythmia initiation.

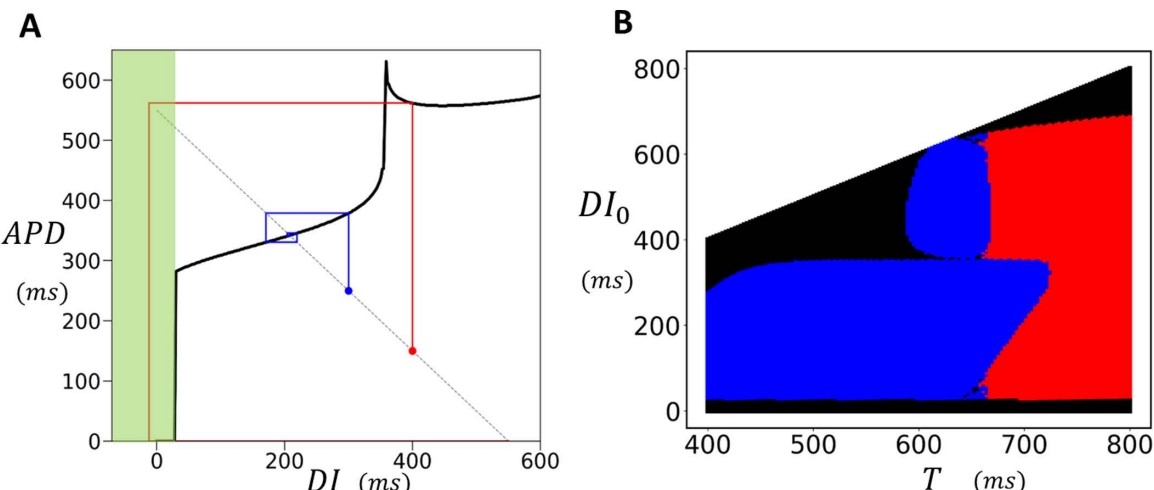

**Fig 9. Conduction block in Cardiac Tissue.** (A) S1S2 restitution curve for a tissue strip $60 \times 3$ cells. All cells on a strip of size $10 \times 3$ are stimulated simultenously using an S1S2 protocol and the APD is measured at the distal end at cell [2,60]. Black line denotes a cubic splines fit to the S1S2 restitution curve. Conduction block occurs for S2 intervals shorter than approximately $DI \approx 25ms$ (green shaded region). Cobweb diagrams are drawn using a cubic spline fit to the restitution curve for initial conditions $DI_0 = 300ms$ (blue line) and $D_0 = 400ms$ (red line). Here, the red line indicates a sequence leading to conduction block. (B) Phase diagram of the system using the cubic splines fit to the S1S2 restitution curve. The y-axis represents the initial diastolic interval $DI_0$, while the x-axis represents the pacing period $T$. For each initial condition, the nonlinear map was iterated 100 times. In the resulting phase diagram, blue regions indicate a stable period-1 solution, red regions indicate a stable period-2 solution, and black regions represent conduction block, i.e., the system reaches $DI < 25ms$ at some point during the 100 iteration trajectory.

## Conclusion

In conclusion, our study provides a comprehensive study of EADs and their effects at both the cellular and tissue levels. We find that EADs in isolated cells are highly variable due to subcellular stochastic processes, leading to significant beat-to-beat variability in the APD. However, in electrically coupled cardiac tissue, these fluctuations are largely dampened, and the dynamics can be described by a deterministic model. Our main finding is that EADs in tissue manifest as alternans, where EADs occur on every other beat. This transition to EAD alternans is discontinuous and leads to pronounced spatial heterogeneities which are strongly arrhythmogenic. These findings provide critical insights into the mechanisms by which cellular stochasticity can cause tissue-level arrhythmogenic events, emphasizing the importance of both nonlinear dynamics and electrical coupling in arrhythmia onset and propagation.

## Supporting information

**S1 Movie. Wave Propagation in 2D Cardiac Tissue with EADs.** This movie shows the full evolution of planar wave propagation in a 100 × 100 cardiac tissue, paced along a 10-cell-wide strip on the left edge. Initially, the tissue is paced for 20 beats at a cycle length (CL) of 420 ms, where the cell model does not exhibit alternans. The CL is then switched to 530 ms, where early afterdepolarizations (EADs) emerge in the deterministic limit. The movie illustrates the progressive development of heterogeneities in the action potential (AP) wave back, culminating in wave break and the formation of reentrant circuits by the 43rd beat. The snapshots shown in the main text are taken from this movie
(MP4)

**S1 Table. Diffusion time scales.**
(DOCX)

**S1 Text. Calcium cycling model equations.**
(DOCX)

**S2 Table. Ca cycling flux parameters.**
(DOCX)

**S3 Table. Spark rate parameters.**
(DOCX)

**S4 Table. Constant parameters.**
(DOCX)

## Author contributions

**Conceptualization:** yohannes shiferaw.

**Formal analysis:** yohannes shiferaw.

**Funding acquisition:** yohannes shiferaw.

**Investigation:** Jack Stein, d'artagnan greene, Flavio Fenton, yohannes shiferaw.

**Methodology:** yohannes shiferaw.

**Project administration:** yohannes shiferaw.

**Resources:** yohannes shiferaw.

**Supervision:** d'artagnan greene, Flavio Fenton, yohannes shiferaw.

**Validation:** Jack Stein, d'artagnan greene, yohannes shiferaw.

**Visualization:** yohannes shiferaw.

**Writing – original draft:** d'artagnan greene, Flavio Fenton, yohannes shiferaw.

**Writing – review & editing:** d'artagnan greene, Flavio Fenton, yohannes shiferaw.

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
