## [Decision Letter · Decision Letter 0]

2 Feb 2025

PCOMPBIOL-D-24-01974

Mechanism of Arrhythmogenesis Driven by Early After Depolarizations in Cardiac Tissue

PLOS Computational Biology

Dear Dr. shiferaw,

Thank you for submitting your manuscript to PLOS Computational Biology. After careful consideration, we feel that it has merit but does not fully meet PLOS Computational Biology's publication criteria as it currently stands. Therefore, we invite you to submit a revised version of the manuscript that addresses the points raised during the review process.

Please submit your revised manuscript within 60 days Apr 04 2025 11:59PM. If you will need more time than this to complete your revisions, please reply to this message or contact the journal office at ploscompbiol@plos.org. Please include the following items when submitting your revised manuscript:

We look forward to receiving your revised manuscript.

Kind regards,

Alison Marsden

Academic Editor

PLOS Computational Biology

Jason Haugh

Section Editor

PLOS Computational Biology

**Journal Requirements:**

**Reviewers' comments:**

Reviewer #1: In this study, Stein and colleagues investigate the mechanisms underlying early afterdepolarizations (EADs) in cardiac tissue - first developing a phenomenological stochastic model that reproduces action potential duration (APD) variability, and then coupling cells in a 2D tissue. The authors demonstrate that the 2D tissue dynamics reproduce the deterministic limit of the stochastic cell cell model, which produces EAD alternans.

Overall, this is a rigorous and well-written study. I have only a few minor comments for the authors to address.

1. Clarification: For the 2D tissue data shown in Figure 4 and subsequent figures, does this also incorporate the reduced LCC inactivation, as in the single cell data shown in Figure 3? It seems as if this is the case, but not explicitly stated.

Similarly, for the statistics (distributions and standard deviation measure) in tissue, do these measures include values for all points in space and in time? Or is APD first averaged across the tissue and then distribution based on time? Please clarify as applicable.

2. Following the data in Figure 6C, D, how wide is the regime in which the "intermittent fluctuations" persist? I.e. How wide is the range between small fluctuations (shown in Figure 6A, B) and alternans (shown in Figure 6E, F)? It would be important to know if this behavior is fairly robust, or a consequence of transitioning between these regimes.

Additionally, the maximum spatial gradient may be another useful metric to quantify for these spatial simulations. It would be interesting to see if the spatial gradients are larger in the alternans regime or the intermittent fluctuations regime.

3. The demonstration of stable spatially discordant alternans in Figure 6F is interesting. It would be beneficial to discuss how this result at slow pacing rates differs from the more classical demonstration of spatially discordant alternans that arises at faster pacing rates and engages conduction velocity (CV) restitution. Presumably CV restitution is minimally engaged at these slow cycle lengths.

4. Minor: The authors state that the wavebreaks shown in Figure 7 is robust and reliably occurs across multiple runs. Please provide an 'n' for the number of trials.

5. The authors nicely comment on how the diffusion coefficient effectively determines the length scale of spatial coupling. Can the authors add to the discussion a bit to also highlight how reduced gap junctional coupling would also be expected to impact these results?

Reviewer #2: This study by Stein et al. investigates Early-after depolarizations (EADs) and their role in cardiac arrhythmia.

Main results: Using a computational model, the researchers found that EADs, which vary randomly in single cells, become more synchronized in cardiac tissue due to gap junction coupling. They discovered that EADs emerge after a sudden transition to alternans, leading to synchronized EADs on every other beat. This transition, caused by a small change in pacing rate, is highly arrhythmogenic, promoting conduction block and reentry. The study highlights the critical role of EAD alternans in arrhythmogenesis, suggesting that ectopic beats are not necessary.

The manuscript is well-written, with a clear and commendable writing style. The discussion section is detailed, very interesting to read, and provides valuable insights into the study's implications.

The scope and findings of the study align well with the journal's focus, and the reviewer believes that the manuscript will meet the high-quality standards expected by "PLOS Computational Biology" if a couple of concerns are addressed:

1) "has been modified for the ventricular cell geometry"

Has the model been modified only for cell geometry? Atrial cellular electrophysiology exhibits significantly different characteristics and may require distinct adjustments.

2) The (numerical) setup for the 2D simulations could be discussed in more detail in the methods section.

3) In the discussion, the authors address the length scale of electrical coupling and the differences that arise between 1D and 3D tissue.

However, the simulations in this study were conducted only in 2D tissue.

I guess it would be feasible to perform simulations on a realistic 3D tissue patch, which would align more closely with the focus of this part of the discussion.

Open-source simulation frameworks such as openCMISS, openCARP, and Chaste are available and could facilitate such an approach.

Do the authors anticipate any differences in their other findings if a proper 3D (finite element) tissue model was used instead of their 2D tissue representation?

4) While the manuscript and the "Online Supplement" provide many equations and model parameters, I believe it would be challenging to fully replicate the results of this study, particularly the 2D setup.

Hence, I would not agree with the statement in the submitted file that

"All relevant data are within the manuscript and its Supporting Information files."

Notably, the computational code is neither mentioned nor made available in the study, which however seems to be required for submissions in PLOS Computational Biology:

"From the time of publication, Authors are required to make fully available and without restriction all data and computational code underlying their findings."

At a minimum, providing e.g. a CellML model implementation, like it is available for the mentioned Mahajan et al. model (https://models.physiomeproject.org/exposure/5a23665da1793b4bfbad92defe7e639d), would be necessary to meet these requirements and ensure reproducibility.

Further, for computational studies like this it would be very helpful for reviewers to have a code framework available to repeat the simulations.

It is well known that there could be a large variability of results depending on choices on numerical methods, see e.g. works of Pathmanathan et al. ( https://doi.org/10.1002/cnm.2467) or Creswell et al. (https://doi.org/10.1098/rsif.2023.0369).

The equations presented in this paper appear sound. However, it is not possible for this reviewer to evaluate whether the chosen numerical framework is also appropriate. Thus it is challenging to assess if the main results mentioned above are justified.

Reviewer #3: The study is interesting, potentially very important, well-conducted, explained in detail and with substantial analysis and context. I don’t have any major concerns about the key results or the analyses but do have some mechanistic questions and some minor issues that would need addressing.

1. Line numbers would always be helpful to identify specific statements.

2. Definition of EADs in the introduction omits any indication of a second upstroke in the AP. Are EADs in this case considered generally as prolonged APD (without or without an upward inflection)? Figures illustrate that the EADs do contain an upstroke, so it may be more concise and accurate for the definition of EADs to reflect this.

3. I think it could be made clearer in the introduction the mechanism by which the stochasticity in calcium handling underpins EADs. Because it is debated whether a spontaneous release during an AP can directly elicit an EAD, which might be the obvious mechanism that comes to the reader’s mind, but this is not the mechanism investigated in this study, it would benefit to be clear about how the noise/stochasticity could underlie APD changes which then can lead to EADs by an ICaL mechanism (rather than a Jrel/NCX mechanism).

3b. Similarly, I think it would be beneficial in the results section somewhere to directly illustrate the mechanism by which noise -> APD variation -> bifurcation of yes/no EADs using the single cell model results. (e.g., showing ICaL traces and how the noise impacts these).

4. I very much support the strategy to deal with LTCC inactivation differences depending on whether the channels are adjacent to sites of large local Ca or not.

5. Is the 2D tissue model isotropic or anisotropic? If isotropic, then could anisotropic coupling, where rather than an average it is comprised of strong (longitudinal) and weak (transverse) coupling, affect the results due to the different interactions of electrotonic load?

6. I would like to see the statement of conduction velocities associated with the diffusion parameters used in the model (and how the CV varies with this variation in parameters and CL) so that we can better judge the electrotonic load conditions.

7. I support the classification of APD at a voltage threshold to enable the consistent analysis in these complex situations. I suggest that the term “APD-40mV” could be used, analogous to “APD90” etc, so that it is always crystal clear which APD is being referred to, and to enable easy interpretation by readers who may miss this definition.

8. The authors discuss how their results regarding 1D strands are relevant for Purkinje induced arrhythmia, and this is certainly true and relevant. However, I think it is also worth noting that conditions such as increased fibrosis, connexin remodelling, ischemia etc could all lead to reduced coupling in tissue, including situations where 2D/3D tissue behaves more like loosely connected 1D strands, which would therefore have the same functional implications. This provides a more general context for how these mechanisms could be relevant in a range of disease conditions and may become more prominent during structural remodelling compared to health.

8b. Moreover, recent studies have highlighted other coupling mechanisms (e.g., ephaptic coupling) that could reduce electrotonic load during repolarisation, which may more readily enable EADs to appear (for example, this mechanism has been shown to more readily enable spatially discordant APD alternans). It may be worth mentioning this in the discussion in regards to the possibility that this present study underestimates the impact of EADs in 3D due to this overly high electrotonic load, and therefore that the mechanisms observed here may be even more relevant in general than is implied by the manuscript.

9. In Figure 3, it could be very helpful to colour the corresponding AP traces in B using the red/black colour scheme used in A and C.

10. In Figure 6, I suggest that the scales of the y-axes in the left and right panels are set to be fully consistent, such that the ticks are aligned and that the scale of the variation in the right-hand panels visually directly corresponds to that shown in the left-hand panels. I may also suggest to add the full label “beat n” rather than just “n” on the x-axis of the left panels.

11. A supporting information video of the case that led to re-entry would be very helpful to fully see this mechanism in action. At least, some more temporal snapshots within the paced beat that induces the re-entry would good. In general, it could be beneficial to show more 2D tissue panels that illustrate the spatial distribution of EADs (are they localised, whole tissue, centre of tissue only?)

**Have the authors made all data and (if applicable) computational code underlying the findings in their manuscript fully available?**

Reviewer #1: **No: ** No code is provided or referenced as being publicly available.

Reviewer #2: **No: ** see Comments to the Editor and the Authors.

Reviewer #3: **No: ** Data are all shown but I could not see a link to model code. Perhaps I missed it.

PLOS authors have the option to publish the peer review history of their article (what does this mean? ). If published, this will include your full peer review and any attached files.

**Do you want your identity to be public for this peer review?** For information about this choice, including consent withdrawal, please see our Privacy Policy .

Reviewer #1: **Yes: ** Seth Weinberg

Reviewer #2: No

Reviewer #3: No

**Figure resubmission:**
---

## [Decision Letter · Decision Letter 1]

1 Apr 2025

Dear Mr shiferaw,

We are pleased to inform you that your manuscript 'Mechanism of Arrhythmogenesis Driven by Early After Depolarizations in Cardiac Tissue' has been provisionally accepted for publication in PLOS Computational Biology.

Best regards,

Alison Marsden

Academic Editor

PLOS Computational Biology

Jason Haugh

Section Editor

PLOS Computational Biology

Reviewer's Responses to Questions

**Comments to the Authors:**

Reviewer #1: The authors have addressed all of my concerns. Congrats on an excellent study.

Reviewer #2: The authors have addressed all of my concerns.

I was able to download, compile, and execute the provided Fortran code successfully.

Additionally, it worked seamlessly with gfortran, which is widely available across all operating systems.

Reviewer #3: I am satisfied with the revisions and the responses to all of my comments and would like to congratulate the authors on an excellent paper.

I only have one small comment that the authors may choose to address: That is indeed a very low conduction velocity. Mechanistically, this is not an issue really because everything scales with WL relative to tissue size. However, I think it is worth a mention in the limitations, sensibly in the new section line 345-356 onwards about sensitivity to these parameters, that this is a small tissue size and a conduction velocity below what is expected, leading to a squareness in the spiral waves.

**Have the authors made all data and (if applicable) computational code underlying the findings in their manuscript fully available?**

Reviewer #1: Yes

Reviewer #2: Yes

Reviewer #3: Yes

PLOS authors have the option to publish the peer review history of their article (what does this mean? ). If published, this will include your full peer review and any attached files.

**Do you want your identity to be public for this peer review?** For information about this choice, including consent withdrawal, please see our Privacy Policy .

Reviewer #1: **Yes: ** Seth H. Weinberg

Reviewer #2: No

Reviewer #3: No

---

## [Editor Report · Acceptance letter]

PCOMPBIOL-D-24-01974R1

Mechanism of Arrhythmogenesis Driven by Early After Depolarizations in Cardiac Tissue

Dear Dr shiferaw,

I am pleased to inform you that your manuscript has been formally accepted for publication in PLOS Computational Biology. Your manuscript is now with our production department and you will be notified of the publication date in due course.

With kind regards,

Anita Estes
